# Integrating Produce Prescriptions into the Healthcare System: Perspectives from Key Stakeholders

**DOI:** 10.3390/ijerph191711010

**Published:** 2022-09-02

**Authors:** Alyssa Auvinen, Mary Simock, Alyssa Moran

**Affiliations:** 1Washington State Department of Health, Tumwater, WA 98501, USA; 2Department of Health Policy and Management, Johns Hopkins Bloomberg School of Public Health, Baltimore, MD 21205, USA

**Keywords:** produce prescriptions, chronic disease, nutrition, food insecurity, healthcare system

## Abstract

People with low incomes suffer disproportionately from diet-related chronic diseases and may have fewer resources to manage their diseases. The “food as medicine” movement encourages healthcare systems to address these inequities while controlling escalating healthcare costs by integrating interventions such as produce prescriptions, in which healthcare providers distribute benefits for fruit and vegetable purchases. The purpose of this study was to identify perceived facilitators and barriers for designing and implementing produce prescriptions within the healthcare system. Nineteen semi-structured in-depth interviews were conducted with experts, and interviews were analyzed using thematic analysis. Overall, interviewees perceived that produce prescriptions could impact patients’ diets, food security, disease management, and engagement with the healthcare system, while reducing healthcare costs. Making produce prescriptions convenient to use for patients, while providing resources to program implementers and balancing the priorities of payers, will facilitate program implementation. Integrating produce prescriptions into the healthcare system is feasible but requires program administrators to address implementation barriers such as cost and align complex technology systems (i.e., electronic medical records and benefit/payment processing). Engaging patients, clinics, retailers, and payers in the design phase can improve patient experience with a produce-prescription program; enhance clinic and retail processes enrolling patients and redeeming benefits; and ensure payers can measure outcomes of interest.

## 1. Introduction

Poor diet is the leading cause of premature mortality and cardiometabolic diseases including heart disease and type-2 diabetes [1]. People with low incomes suffer disproportionately from these chronic conditions [2] and may have fewer resources to manage their diseases due to the high cost of healthful foods [3] and psychosocial impacts of food insecurity [4]. Federal nutrition assistance programs, which provide targeted benefits to millions of low-income Americans to purchase food, alleviate food insecurity [5,6]; however, these supports are often inadequate to meet key dietary guidelines [7]. For people managing diet-related chronic conditions, poor diet quality exacerbates health inequities by increasing incidence of disease complications. These complications have been shown to spike when financial resources are scarce, increasing hospitalizations [8] and emergency-room visits [9] and leading to higher healthcare costs [10].

The “food as medicine” movement encourages healthcare systems to address these inequities while controlling escalating healthcare costs by integrating healthy, medically appropriate foods into a patient’s treatment plan [11]. Produce-prescription programs are one promising food-as-medicine approach, which allows healthcare providers or health insurance payers to distribute targeted “cash” benefits to eligible patients to buy produce [12]. Research on the health effects of produce prescriptions is nascent [13], with many prior studies using observational, uncontrolled study designs. However, some of this work has shown promising results such as increased fruit and vegetable consumption among children [14], improved food security for households with children [15], increased fruit and vegetable consumption among hypertensive adults [16], and better blood glucose control among patients with a history of uncontrolled diabetes [17]. Evidence from similar programs has shown positive effects of financial incentives on dietary behaviors, including in studies using randomized controlled experimental designs [18,19,20,21].

Current produce-prescription programs have limited reach and sustainability due to piecemeal funding from relatively short-term grants. For example, the Agricultural Improvement Act of 2018 (Farm Bill) authorized the Produce Prescription Program, which provides grants to organizations to implement and evaluate produce prescriptions, with the goal of improving dietary behaviors, reducing household food insecurity, and reducing healthcare use and associated costs [22]. However, the grants only provide USD 500,000 for up to three years for organizations to implement and evaluate their produce-prescription programs; 29 programs were funded across the U.S. from 2019–2021 [23]. Implementing produce-prescription programs in collaboration with healthcare providers and payers, specifically Medicaid, could expand the reach of these services due to the size of the Medicaid population (more than 83 million people in 2021) [24], ensure sustained funding, and be cost-effective in the long-term [25]. However, there are major gaps in our knowledge about how produce prescriptions could be implemented in this context.

The goal of this research is to provide guidance from key stakeholders on challenges and opportunities to integrate produce prescriptions into the healthcare system and state Medicaid plans. Through in-depth key-informant interviews, we answer the following research question: What are the perceived facilitators and barriers for designing and implementing produce prescriptions within the healthcare system? In this study, Medicaid was used as an example payer on which to base the interview questions, but informants provided information that may be applicable to a broad range of healthcare payers.

## 2. Materials and Methods

### 2.1. Study Sample

Interview participants were identified using purposive sampling. Researchers identified nine informant categories based on subject-matter expertise and experience implementing and planning produce-prescription programs that are connected to Medicaid (Table 1). Informant categories were selected based on the ability of interviewees to provide expert insight into design, implementation, or evaluation of produce prescriptions. Produce-prescription program administrators (e.g., a program manager at a community-based organization (CBO)), clinicians (e.g., dietitian), and food retailers (e.g., a store marketing director) were identified as important stakeholders because they have prior experience administering produce-prescription programs and can describe best practices and barriers in clinical and retail processes and systems for scaling these programs. Healthcare law (e.g., a food-policy lawyer), Medicaid managed care (e.g., a director of community programs), and government stakeholders (e.g., a health policy analyst) were selected to provide expertise on potential funding mechanisms, legal and policy considerations, and staffing capacity and other resources for program implementation. Academic research stakeholders (e.g., a university professor) were selected for their expertise in program evaluation, study design, and measurement of outcomes related to produce prescriptions. Advocacy stakeholders (e.g., an executive director at a non-profit organization) provided insight into the influence of political will and context on scaling produce prescriptions through Medicaid. A technology provider (e.g., an electronic benefit transfer (EBT) processor) was also identified as an informant category to provide expertise on payment processing, however, the two contacts identified for this study did not accept an interview, and this area of expertise was filled by the food retail and produce-prescription program administrator stakeholders. Twelve contacts were identified based on the research team’s subject-matter expertise, and all 12 contacts accepted an interview. An additional nine contacts were identified through expert recommendation, seven of whom accepted an interview. A total of 19 participants were included in the research.

### 2.2. Data Collection

Interviews were conducted by one researcher (AA) by web conference or over the telephone. Interview questions were asked in accordance with a semi-structured interview guide and adapted to each interviewee’s area of expertise. Two researchers (AA/AM) developed the interview guide (Appendix A), which was tested for content validity and piloted for clarity with one clinician and academic expert in food-as-medicine interventions in July 2020. The interview guide was revised based on feedback from the pilot interview. Interviews were conducted from July–October 2020. Nineteen interviews were conducted in English and lasted between 25 and 75 min. Eighteen of the nineteen interviews were audio recorded and transcribed verbatim. One interviewee did not wish to be recorded and so the researcher took detailed notes. Interviewees were asked their perceptions of the benefits and challenges of implementing produce-prescription programs through Medicaid; organizational alignment (or not) with produce prescriptions; and perceptions regarding how a program should be designed, implemented, evaluated, and funded within the context of a state Medicaid program. Some questions were tailored to the respondent’s expertise. For example, the lawyer was asked what specific Medicaid policy mechanisms were available to support and scale produce prescriptions and clinicians were asked to describe how clinics could administer a produce-prescription program (e.g., what resources are needed at the clinic-level?). This study was deemed non-human subjects research by the Johns Hopkins Bloomberg School of Public Health Institutional Review Board on 4 June 2020.

### 2.3. Data Analysis

One researcher developed the analytic codebook using an inductive–deductive approach, in which emergent codes were created and then mapped onto constructs from the Consolidated Framework for Implementation Research (CFIR), when applicable [26]. CFIR is a theoretical model often used in implementation research and was selected for this study to elicit information on barriers and facilitators to implementation. CFIR provides a menu of constructs arranged across five domains that have been associated with effective implementation of programs and policies (intervention characteristics, outer setting, inner setting, characteristics of individuals, and process) [26]. To develop the codebook, one researcher (AA) read five transcripts and drafted an initial codebook, which was shared with a second researcher. The two researchers then coded two transcripts independently, assessed inter-rater agreement, discussed disagreements, and refined the codebook until agreement was reached. The final codebook included 63 codes that mapped onto 12 CFIR constructs. Both researchers independently coded the 17 remaining transcripts.

Coded transcripts were analyzed by one researcher (AA) using thematic analysis. This involved identifying patterns within the data by grouping codes and analyzing relationships between them. Emergent themes were mapped onto the CFIR framework, when appropriate, and subthemes were identified. Ten themes were identified within the domains of intervention characteristics (relative advantage, adaptability, complexity, cost, evidence strength and quality), outer setting (patient needs and resources, external policies and incentives), inner setting (implementation climate, readiness for implementation), and process (planning). There were no themes tied to the characteristics of individuals domain. Quotations were selected to illustrate findings and provide depth. Data were analyzed using ATLAS.ti, version 8.

## 3. Results

In total, 19 key informants participated in the study. Informants discussed a variety of facilitators and barriers to consider when designing, implementing, and evaluating a produce-prescription program that is integrated in the healthcare system. The following summarizes 10 themes that emerged from interviews. These themes are presented in Table 2 according to CFIR domains and select representative quotes are included.

### 3.1. Facilitators

**Better patient engagement, improved health outcomes, and decreased healthcare costs**. Produce prescriptions may improve patient engagement, satisfaction with care, and trust in healthcare providers. One payer representative described improved connections between primary care providers and patients as a result of a produce-prescription program, and that increased patient engagement was a driving factor for the payer agreeing to partner with a community-based organization for further iterations of the program. Another payer representative noted that produce prescriptions are one way to help patients who have been disenfranchised from the healthcare system to reconnect and build trust with the healthcare system and providers. Clinicians and produce-prescription administrators described better patient engagement as one factor contributing to why clinics offer produce prescriptions to their patients. Many respondents perceived that produce-prescription programs may lead to better diet quality, health outcomes, and quality of life for patients who are food insecure or have a diet-related chronic condition. Payers and clinicians acknowledged it can be difficult for patients to prioritize purchasing healthy food in the presence of other financial stressors, and that produce prescriptions could help address financial needs. One researcher described that the potential impact of produce prescriptions on healthcare utilization and costs may be greatest for patients with a combination of clinical and social complexities (e.g., a person who has difficulty managing a diet-sensitive illness and is unable to afford healthy foods).

**Using patient eligibility criteria to balance program goals and costs.** Tying program participation or patient eligibility back to a health condition (e.g., pre-diabetes, diabetes, heart disease) was generally viewed by respondents as important, particularly if programs are publicly funded. Overall, there was a sense that produce-prescription programs could serve a wide range of patients, and food insecurity is a logical eligibility criterion. Views varied widely across respondents about other eligibility criteria—some respondents thought the target population should be limited to those with acute needs (e.g., difficulty managing a diet-related chronic disease), whereas other respondents thought produce-prescriptions programs could be offered preventively for people at risk of developing a diet-related chronic disease or for pregnant women or parenting families as a means to establish healthy habits early in life. Many respondents said that the target population and the subsequent health outcomes to track should be dictated by the healthcare providers or healthcare payer. For example, if a healthcare payer is paying for produce prescriptions for their members with diabetes, changes in hemoglobin A1c are important to measure.


**Engaging clinics, retailers, and participants in program design.**


Clinics and retailers were mentioned by many of the respondents as critical to engage in the planning and design phase of a produce-prescription program because clinical enrollment or referrals (e.g., screening for food insecurity, distributing the produce prescription, etc.) and payment processing (e.g., scanning a voucher or recognizing a card benefit) are two key components to make sure a program functions properly. Produce program administrators specifically noted that retailers or payment processors are important to involve in the development of a produce-prescription program, as retail systems will dictate the steps a patient must follow to redeem the produce prescription—which can ultimately make the program easy or difficult for the patient to use. Including patients in the program design was mentioned by less than half of respondents. These respondents—which included a clinician, government stakeholder, program administrator, researcher, lawyer, and two payers—thought including patients in program design was very important but recognized it is difficult to do.


**Making produce prescriptions accessible and relevant to patients’ lives.**


Ensuring produce-prescription programs are convenient for patients was brought up in a few different contexts, including: ensuring that the prescription can be used at a variety of retail outlets; allowing prescriptions to be used for online grocery shopping and delivery; allowing the benefit to be used across more than one shopping trip; and providing clear instructions on how to use the benefits. One program administrator mentioned that they are making changes to their produce-prescription program to offer online ordering and grocery delivery, which may better meet the needs of patients who are homebound. The same program administrator described that, by transitioning their program from a paper-based voucher to a card-based benefit, the patient has the flexibility to use their benefits to best suit their shopping and eating habits. For example, with a card-based benefit, people can spend part of their benefit in one shopping trip and the rest at a later time, rather than being required to spend the entire benefit in one transaction (the “use it or lose it” model). One clinician described using videos and texting patients to make sure they understood how the produce-prescription program worked. Four respondents described the importance of cultural relevance, for example, by ensuring culturally relevant foods are available through produce-prescription programs.

**Providing resources to clinics and retailers.** Equipping clinics and retailers with clear instructions and resources for implementing a produce-prescription program was mentioned by more than half of respondents. One researcher suggested that healthcare payers should develop an implementation toolkit for healthcare systems, which could include appropriate referral and billing instructions. An advocate mentioned that training clinical staff on produce prescriptions would need to be monitored, for example, through a continuous plan–do–study–act cycle, to alleviate workflow issues that may arise. Training for retail staff was mentioned by all of the retailers and clinicians interviewed. One retail respondent described how cashiers can help program participants maximize their food benefits with a verbal prompt at the cash register (e.g., “Did you know you still have $5 left for fresh fruits and vegetables?”).

**Using state-level Section 1115 Demonstration Waivers and Medicaid managed care for funding models.** Respondents described mechanisms to pay for produce prescriptions with Medicaid dollars would vary by state, and a government respondent, healthcare payer, and lawyer, mentioned that state Medicaid agencies have or are looking to the Centers for Medicare and Medicaid Services (CMS) to see if paying for produce prescriptions with Medicaid funds is acceptable. Seven of the respondents described the use, or potential use, of a state’s Section 1115 Demonstration Waiver to pay for produce prescriptions among the Medicaid population. However, one healthcare payer noted that sustained funding for produce prescriptions through Medicaid should be considered as a standard Medicaid benefit that is not waiver-driven because waivers tend to “go away”. Program administrators, government stakeholders, payers, and a lawyer described working with Medicaid managed care organizations (MCOs) to fund produce prescriptions within the Medicaid population. The lawyer interviewed for this study described three primary ways for MCOs to use Medicaid funds to pay for produce prescriptions: (1) in lieu of services—in which state Medicaid programs can cover non-medical services to prevent incurring more expensive medical services in the future (e.g., produce can be covered with the justification that a healthful diet can prevent a more serious chronic condition); (2) as a value-added service—in which a service is offered to enrollees; or (3) as a quality improvement activity—which includes activities that improve healthcare quality. Outside of direct funding from Medicaid, one respondent noted that, instead of building produce prescriptions into Medicaid managed care contracts, a state could appropriate funding to MCOs to pilot produce prescriptions, which could, in turn, provide evidence for state Medicaid agencies to build produce prescriptions into Section 1115 Demonstration Waivers or into MCO contracts.

### 3.2. Barriers

**Improving the evidence for produce prescriptions.** Respondents from every informant category mentioned the importance and/or difficulty of evaluating the effectiveness of produce-prescription programs. One advocate and one payer described the need for longitudinal data to determine if healthcare engagement, health behaviors, and health outcomes are impacted as a result of produce prescriptions. The payer described the importance of being able to tie patient clinical data with retail point-of-sale data for produce-prescription redemption, both in the formative stages of a program and to determine if the program is affecting desired outcomes. However, that same payer recognized that linking data across clinical and retail systems can be difficult. An advocate also described the need for biological readings and case-control studies to determine that produce prescriptions are effective; from the advocacy point-of-view, this information would significantly bolster the evidence for produce prescriptions.

**Buy in and capacity.** Clinicians, payers, government stakeholders, retailers, and program administrators described the need for leadership buy in and staff/resource capacity across multiple settings. One clinician described the popularity of their produce-prescription program among staff who are carrying out direct patient interaction (e.g., care managers, community health workers, and dietitians). Buy in to implement produce prescriptions at the clinic leadership level was mentioned by clinicians and government stakeholders. However, one payer noted that staffing capacity, not political will, was a primary challenge for produce-prescription implementation. The payer described working with different clinics to enroll patients into a produce-prescription program and how staffing capacity varied among those clinics. That same payer described using their organization’s community health workers to assist with patient enrollment, although these workers also had limited time to participate.

**Integrating with existing technology systems.** Respondents from all informant categories described the importance and difficulty of tying produce prescriptions to existing technology systems. Clinicians, program administrators, government stakeholders, advocates, a lawyer, and a researcher described using an electronic medical record (EMR) as an important implementation tool for produce prescriptions. For example, the EMR can prompt clinicians to screen patients for produce prescription eligibility, directly enroll or refer patients into a produce-prescription program, and track patient outcomes. Several respondents familiar with clinical workflows noted that the ability to track utilization of the produce prescription through the EMR would be especially useful, because logging into a separate platform to view benefit redemption information is disruptive to their typical clinical workflow. Several respondents recognized that embedding produce prescriptions into an EMR would require healthcare systems to re-program their EMR, which is time-consuming and difficult to do. One produce-prescription administrator, who works with a clinic and pharmacy to implement a produce-prescription program, identified the lack of an International Classification of Diseases (ICD)-10 code as a barrier to writing prescriptions. Typically, an ICD-10 code would be used within an EMR to send an e-prescription to the pharmacy, but there is no ICD-10 code for produce. The program administrator described how the produce prescription is created and tracked manually, which negatively impacts clinic and pharmacy staff time: the clinician works with a care coordinator to fax a produce prescription over to the pharmacy, the pharmacy receives the fax, tracks produce prescriptions in a spreadsheet, and then distributes the prescription to the patient based on their spreadsheet records. Respondents also expressed challenges for benefit delivery and redemption. Retailers described the need to streamline benefit redemption (e.g., use an electronic/card-based benefit versus paper vouchers or tokens). Program administrators also acknowledged a need for technology improvements in benefit delivery and redemption, which would allow patients to use their produce prescriptions across multiple retail settings.

**Increased costs across organizations.** Respondents noted that making system changes to EMRs, benefit delivery systems, and retail point-of-sale software may be prohibitively expensive. While the costs for systems changes may be expensive upfront, one program administrator noted that implementing system changes (e.g., efficient payment processing) allows a program to scale and would ultimately minimize the administrative costs relative to the number of patients served by the program. Other than the cost of the incentives and system changes, respondents described the costs of increased staff time to administer and manage produce-prescription programs. At the clinic level, staff time was needed for patient enrollment and program-related follow-up. One clinician described enrolling patients into the produce-prescription program during patients’ normal nutrition counseling visit (i.e., not using a separate visit to enroll the patient into the program), without negatively impacting clinician productivity. A few respondents described the need for payers to dedicate administrative staff time to produce-prescription programs. For example, a staff member could scan claims data to identify eligible patients for the program.

## 4. Discussion

This study identified perceived facilitators and barriers for designing, implementing, and evaluating produce prescriptions within a state Medicaid program. Researchers conducted 19 interviews with experts in eight key-informant categories, including a produce-prescription administrator, clinician, food retailer, healthcare lawyer, healthcare payer, government stakeholder, academic researcher, and advocate. Overall, integrating produce-prescription programs into a state Medicaid program was viewed positively, with potential to improve patients’ diets, food security, disease management, financial security, and experiences interacting with the healthcare system. Facilitators for incorporating produce prescriptions into a state Medicaid program included factors that impact patients, clinics, and policies: aligning patient eligibility criteria with program goals; ensuring the program is convenient (i.e., easy-to-redeem benefits, includes preferred retailers) and culturally appropriate (i.e., includes preferred foods); engaging clinics, retailers, and patients in program design; providing resources to clinics and retailers for program implementation; and using Section 1115 Demonstration Waivers and Medicaid Managed Care as funding mechanisms. Barriers included a lack of high-quality evidence for produce prescriptions; identifying clinics that are “bought in” and have capacity (i.e., leadership support) to implement the program; the staff time and costs of coordinating across multiple organizations (i.e., clinics, community organizations, retailers); and the complexity of aligning EMRs and benefit processing systems.

Many stakeholders viewed produce prescriptions as positively impacting patient outcomes and experiences within the healthcare system. These findings are consistent with previous studies describing the client perspective [16,27,28]. For example, one evaluation of a produce-prescription program in Washington State found participants reported being better able to manage their health conditions and meet nutrition and diet-related goals [28]. Another evaluation of a produce prescription program in Cuyahoga County, OH found that offering produce prescriptions to patients influenced the patient–provider interactions in positive ways [16]. For example, providers felt better prepared and empowered to advise program participants to eat more fruits and vegetables [16].

Some stakeholders highlighted the importance of including patients in the program design process, which may greatly improve program acceptability among participants. Nutrition assistance programs such as the Special Supplemental Nutrition Program for Women, Infants, and Children (WIC) have struggled with participant-centered design [29]. However, there is a recognition that participant-centered design can improve services and enable participants to make long-lasting and positive behavior changes [30]. For example, the WIC food package was revised to include grain options suitable for various cultures (e.g., corn meal, buckwheat, teff) [31] and both WIC and SNAP have switched from paper vouchers to card-based systems, significantly improving program participation and benefit redemption [32,33].

Stakeholders from all informant categories identified a need to improve the evidence for produce-prescription programs. A 2022 Food is Medicine Research Action Plan shows that, when compared to other food-as-medicine interventions (i.e., medically tailored meals and medically tailored groceries), produce-prescription research is growing, but the wide range of study designs present challenges for interpretation [34]. For example, 27 studies have been published on produce prescriptions, and study designs range from qualitative interviews to a pilot randomized control trial with 128 participants [34]. While conducting randomized control trials that have sufficient sample sizes and power would be the gold standard for evidence generation, this may be difficult to carry out within current funding constraints.

Most respondents talked about elements of equity, although most did not use this terminology. For example, respondents talked about targeting produce-prescription programs toward people who experience food insecurity and/or chronic medical conditions, which may, in turn, reduce economic and health disparities. Respondents also described including patients in the formative stages of a produce-prescription program, which could improve the likelihood that a patient will use and benefit from the program. One payer described produce prescriptions as a mechanism to address social determinants of health, alleviate financial pressure among their patients, and generally improve population health equity.

### 4.1. Implications for Policy and Practice

For produce prescriptions to be integrated into a state Medicaid program, states must obtain approval from CMS. There are several opportunities for doing this, identified by key informants and described in detail by The Center for Health Law and Policy Innovation in a policy scan published in 2020, including through standard Medicaid, Medicaid managed care, Section 1115 Demonstration Waivers, 1915 Waivers (which allow states to provide home and community-based services to individuals), and Dual Eligible Special Needs Plans and Dual Demonstrations (targeted to beneficiaries who qualify for both Medicaid and Medicare) [35]. States could turn to early adopters of these practices for peer support. For example, Section 1115 Demonstration Waivers allow state Medicaid programs to experiment, pilot, and demonstrate projects to better serve their state’s Medicaid population. In Massachusetts, MassHealth’s Section 1115 Demonstration Waiver is paying for health-related nutrition supports such as produce prescriptions for certain members [36]. Likewise, North Carolina’s Department of Health and Human Services plans to fund produce prescriptions for select Medicaid beneficiaries through their Section 1115 Demonstration Waiver’s Healthy Opportunities Pilots [37]. Additionally, state Medicaid agencies could give preference to contracting with MCOs that offer produce prescriptions to their Medicaid members. Although not highlighted in this study, Medicaid funding could apply to accountable care organizations (ACOs) as well. While using Medicaid as a funding mechanism for produce prescriptions is feasible, it could exacerbate national health inequities because not all states have expanded Medicaid coverage to adults with incomes up to 138% of the federal poverty level (i.e., Medicaid expansion).

The costs of implementing produce-prescription programs, including funding for additional staff time and the integration of prescriptions into healthcare and retail IT systems, was cited as a major barrier across informant categories. Interviewees identified several opportunities for covering these costs. For example, MCOs can incentivize clinics to offer produce prescriptions to eligible patients (e.g., tie patient reimbursement to program use) and pay for clinics to embed produce-prescription screening, enrollment/referral, and tracking into their EMRs. One interviewee mentioned that their state Medicaid agency provided some grant funds to organizations to build out technical infrastructure in preparation for the state’s Section 1115 Demonstration Waiver. Other state Medicaid agencies could consider providing funding for capacity building among clinics, retailers, and program administrators as well as within EMRs and payment processing systems. State Medicaid agencies could use state appropriated funding to provide capacity-building grants to clinics and program administrators or designate a portion of Section 1115 Demonstration Waivers to support capacity building. Although not mentioned by interviewees, funding for capacity building could be prioritized to clinics in areas of high need (e.g., federally qualified health centers that serve majority low-income populations) and that currently lack systems to support food-as-medicine interventions (e.g., outdated EMRs). Prioritizing funds for program administrators with a successful track record of implementing food-as-medicine interventions that could scale produce-prescription distribution (e.g., by integrating benefits onto an EBT card) may help the intervention reach many patients.

Many stakeholders recommended the use of benefit transfer cards to improve participant experience, reduce administrative costs, and allow for better tracking of prescription redemption. Along with cards, produce-prescription programs could expand payment options to include mobile payments, which may make benefits easier to manage, track, and redeem for some participants. Program administrators could examine ways to improve and scale produce-prescription programs by looking to similar fruit and vegetable interventions that have been implemented at scale and for a long time. For example, WIC is like produce-prescription programs in that benefits can only be used for specific food items and participants must receive health screenings and nutrition education by WIC clinic staff in order to receive benefits. WIC was mandated to transition from paper vouchers to EBT cards, which can help destigmatize program use, reduce checkout time for participants, and minimize cashier confusion at stores [33].

### 4.2. Strengths and Limitations

This study has several limitations. Although perspectives from a wide variety of subject-matter experts familiar with produce-prescription programs were included, it is small in scope. Furthermore, no interviews were accepted by one informant category (Technology Provider), which leaves a gap in in-depth understanding of how produce prescriptions can be integrated into payment processing. It is also possible that researchers unconsciously influenced results through the way interviews were conducted or analyzed. Researchers used several strategies to mitigate bias by transcribing interviews verbatim, taking detailed notes during interviews and analysis, and having two researchers code all transcripts. Strengths of the study include the variety of subject-matter experts interviewed from across the U.S.

## 5. Conclusions

This study examined strategies for designing and implementing produce prescriptions within the healthcare system and state Medicaid plans. Respondents viewed produce prescriptions positively, with potential to address health equity by improving patients’ diets, food security, disease management, financial security, and experiences with the healthcare system. Engaging patients, clinics, retailers, and payers in the design of a produce-prescription program can improve patient experience with, and use of, the benefit; enhance clinic and retail processes for enrolling, distributing, and redeeming benefits; and ensure payers are able to track outcomes of interest.

## Figures and Tables

**Table 1 ijerph-19-11010-t001:** Description of key-informant sampling.

Informant Category	Description of Expertise Related to Produce Prescriptions	Number of People Interviewed
Produce-prescription administrator	Administers a produce-prescription program that is working with, or plans to work with, a state Medicaid agency, Medicaid managed care organization (MCO), or accountable care organization (ACO). Administration includes setting up parameters of a produce-prescription program, coordinating payment and data collection among clinics and healthcare providers that offer produce prescriptions, and setting up systems to redeem produce prescriptions at retail outlets.	3
Clinician	Provides produce prescriptions to their patients and understands clinic workflow including enrollment procedures for patients into a produce-prescription program, distribution of benefits to patients, clinical data collection and reporting, and reimbursements.	2
Food retail	Oversees redemption of food benefits at retail outlets, including produce prescriptions and/or other food assistance benefits.	4
Healthcare law	Understands federal and state Medicaid statutes and programs that may allow produce prescriptions to operate and be reimbursable through Medicaid funds.	1
Healthcare payer	Serves Medicaid beneficiaries with comprehensive healthcare and social supports and receives funds from state Medicaid programs to cover beneficiaries’ healthcare.	2
Government	Describes state and federal healthcare and public-health policy, program, and funding mechanisms for produce prescriptions.	3
Academic research	Provides expertise in study designs and data-collection mechanisms that should be considered for evaluating produce-prescription programs.	2
Advocacy	Provides insight into the political landscape at the federal and state levels for implementing and scaling produce-prescription programs.	2

**Table 2 ijerph-19-11010-t002:** Description of themes by CFIR domain.

**CFIR Domain**	**Theme**	**Description**	**Barrier or** **Facilitator**	**Illustrative** **Quote**
Intervention Characteristics	*Relative Advantage:* Better patient engagement, improved health outcomes, and decreased healthcare costs.	Clinicians, payers, produce-prescription administrators, and researchers viewed programs as improving patient engagement, satisfaction with care, adherence to treatment, and trust in healthcare providers. Clinicians and payers also described produce prescriptions as a financial support mechanism so patients can make health behavior changes.	Facilitator	*Historically, the group of* *patients [with diabetes that’s poorly controlled] have been hard to get back into the clinic and having produce prescriptions there, they answer their phones. They come into the clinic and they get their labs drawn. They’re a lot more engaged in care and I don’t think it’s just the free food. I think… it’s part of a total way of caring about people.* (Clinician)
Intervention Characteristics	*Adaptability:* Using patient eligibility criteria to balance program goals and costs.	Tying patient eligibility for produce prescriptions to health conditions (e.g., diabetes, pre-diabetes, heart disease) or food-security status was generally viewed as a strategy for reaching the population most likely to benefit, and can be adapted to meet the needs of each implementing site.	Facilitator	*I think that makes sense to line up with a chronic health condition and being able to also hopefully see objective changes in measures of chronic disease, like hemoglobin A1c, blood pressure and body mass index.* (Clinician)
Intervention Characteristics	*Complexity:* Integrating with existing technology systems.	Respondents from all informant categories described integration of produce prescriptions into electronic medical records and/or retail point-of-sale systems as important, but difficult to implement.	Barrier	*The [electronic medical record] capacity and the different provider settings are going to be an issue. What the [electronic medical records] both can collect and what you can extract. You might be able to put in the answer to the screening questions, but you might have a hard time extracting it or you might have a hard time extracting the health outcome data and being able to link it to the prescription utilization information.* (Government stakeholder)
Intervention Characteristics	*Costs:* Increased costs across organizations.	Respondents described costs to clinicians and payers to implement and evaluate produce prescriptions. Costs included staff time, data needs, and technology infrastructure.	Barrier	*Staff time is significant, especially if you are playing telephone tag with [patients who receive the produce prescriptions].* (Clinician).
Intervention Characteristics	*Evidence Strength and Quality:* Improving the evidence for produce prescriptions.	Advocates, produce-prescription administrators, government stakeholders, payers, and lawyers described the need for more rigorous study designs for produce prescriptions.	Barrier	*I think the biggest fault that I’ve seen in almost all [produce prescription] studies, a lot of them ask the right questions, truly, but it’s the evaluation mechanism is too far from biological readings and the second is that there isn’t a case control. And, I think, if you can add those you’d significantly bolster the science around it pretty quickly.* (Advocate)
Outer Setting	*Patient Needs and* *Resources*: Making produce prescriptions accessible and relevant to patients’ lives.	Clinicians, payers, and produce prescription administrators described the need to make produce-prescription programs convenient for patients to use.	Facilitator	*I work with a lot of people who are pretty low literacy and so, the fact—just something simple like they have made a YouTube video explaining how the program works like seems really small, but the fact that I can just text it to patients and they can watch it as many times as they want and it shows them how to use the program is really, really* *helpful.* (Clinician).
Outer Setting	*External Policy and* *Incentives:* Using State-level Section 1115 Demonstration Waivers and MCO contracts for funding models.	Lawyers, payers, produce-prescription administrators, and government stakeholders described the potential to pay for produce prescriptions with state Medicaid Section 1115 Demonstration Waivers, or through contracting mechanisms with Medicaid managed care organizations.	Facilitator	*The big one is 1115 waivers because they are the most flexible. The great thing about 1115 waivers is you can pay for services that would not otherwise be paid under Medicaid. There’s a lot of flexibility and there’s a lot of precedent.* (Lawyer).
Inner Setting	*Implementation* *Climate:* Buy in and capacity.	Clinicians, payers, government stakeholders, retailers, and program administrators described the need for leadership and staff buy in and enthusiasm for produce-prescription programs, as well as the importance of staff and resource capacity to ensure the program was implemented well.	Barrier	*Political will within the agencies or provider agencies didn’t seem to be a real challenge. I think people got it. It was really a capacity issue. Many just didn’t have the capacity to kind of get it done in the same way.* (Payer)
Inner Setting	*Readiness for* *Implementation:* Resources for clinics and retailers.	Clinicians, retailers, researchers, payers, lawyers, government stakeholders and advocates described the need to equip clinics and retailers with resources for program implementation.	Facilitator	*[Providing] a toolkit [to a clinic] for how to implement it in the clinic, including the screening questionnaire. It’s going to include instructions or guidance that will get integrated into the electronic* *medical record.* (Government stakeholder).
Process	*Planning:* Engaging clinics, retailers, and participants in program design.	Clinicians, researchers, government stakeholders, program administrators, lawyers, and payers thought engaging clinics, retailers, and participants in the design phase of a produce-prescription program was important.	Facilitator	*I think it’s hard to design a program like this that’s going to work without giving* *[potential participants] a seat at the table.* (Government stakeholder).

## Data Availability

Not applicable.

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
