# Peer review of "Integrating Produce Prescriptions into the Healthcare System: Perspectives from Key Stakeholders"

_ijerph, 2022, doi:10.3390/ijerph191711010_

Round 1

Reviewer 1 Report

This paper is a fantastic and vital contribution to the produce prescription research field. While the authors note produce prescription research is vast, only 27 studies have been published on this topic.

Their paper offers novel evidence to the field that includes qualitative perspectives stakeholders who are not usually interviewed (i.e., healthcare lawyer, healthcare payers, food retail, etc). The interviews capture the vast array of expertise needed to design, implement, and evaluate a PPRx program proving these programs require a multidisciplinary approach. 

The "best practices" provided by the stakeholders are noteworthy to a wide audience. For example, the three ways for MCOs to use Medicaid dollars to fund PPRx is useful for MCOs, clinicians, policymakers, and advocates. The need for longitudinal data and RCT studies to determine long term impacts of PPRx programs is unlikely to be funded by federal government or private foundations due to the significant cost. Providing a viable pathways for MCOs to use Medicaid dollars not only provides permanent and sustainable funding for programs but also a solution to tracking longitudinal healthy care outcomes. 

While it is difficult to show better patient engagement, improved health outcomes, and decreased healthcare costs, hearing from payer representatives was useful. They provided what is important to them that cannot be measured or assigned a true cost that a PPRx program can provide - increased patient engagement and improved connections between primary care providers and patients especially for patients who have been disenfranchised by the healthcare system.

This paper provides additional and needed evidence to offer MCOs and Medicaid to fully fund a PPRx program. 

Author Response

Thank you for the comment. We have updated this language to state that the produce prescription literature is “growing” (not vast).

Reviewer 2 Report

This is a well-written piece and I have no real points for review. 

I think using the term 'food as medicine' would put off the people you are trying to help and although what we eat is good/bad for our health, it is dangerous to mix food/medicine in that way.

This is why it is so important to have input from the people you are trying to help. What they think of the scheme, how it could be changed to help them use it, whether anything else is needed to help them change their diet.

Why use Medicaid? Why use CFIR framework?

Author Response

Using term “food as medicine”

Thank you for the comment. This is standard terminology in the field, so we have decided to keep the term “food as medicine” in the manuscript. We do agree that having patient input on branding for such a program would be helpful during implementation design.

Why Medicaid?

Thank you for the comment. Medicaid is the largest healthcare payer in the country, so we focused on questions in the interview specifically around Medicaid. We have added a bit more context in the introduction – Medicaid is an important avenue for reaching tens of millions of Americans with health services (estimated 83 million people on Medicaid in federal fiscal year 2021).

Why use CFIR?

Thank you for the comment. CFIR is a framework that has been widely used in public health and healthcare implementation research (as of August 2022, more than 3,500 published studies citing CFIR in PubMed). We have added a bit more context to the Methods section about why CFIR was chosen.